# P2Y_2_R-Mediated PAK1 Activation Is Involved in ESM-1 Overexpression in RT-R-MDA-MB-231 through FoxO1 Regulation

**DOI:** 10.3390/cancers14174124

**Published:** 2022-08-26

**Authors:** Hana Jin, Hye Jung Kim

**Affiliations:** 1Department of Pharmacology, Institute of Health Sciences, College of Medicine, Gyeongsang National University, Jinju 52727, Korea; 2Department of Convergence Medical Science (BK21 Plus), Gyeongsang National University, Jinju 52727, Korea

**Keywords:** ESM-1, P2Y_2_ receptor, FoxO1, PAK1, RT-R-TNBC cells

## Abstract

**Simple Summary:**

Radiotherapy is an important treatment to treat triple-negative breast cancer (TNBC; ER^−^, PR^−^, HER2^−^) patients. However, the radioresistance of BC cells remains the main obstacle to radiotherapy efficacy, which leads to increased tumor relapse and metastasis. Previously, we discovered that endothelial cell-specific molecule-1 (ESM-1) is the most increased gene in radiotherapy-resistant (RT-R)-TNBC cells compared to their parental cells and determined that ESM-1 plays a critical role in the tumorigenesis of RT-R-TNBC cells through the regulation of several genes related to tumor growth, progression and metastasis. Therefore, in this study, we aim to identify the mechanism by which ESM-1 is overexpressed in RT-R-MDA-MB-231 cells. Our results demonstrate, for the first time, that ESM-1 overexpressed in RT-R-MDA-MB-231 cells is regulated through the P2Y_2_R-PAK1-FoxO1 signaling pathway. Our findings suggest that targeting this pathway may be a new therapeutic target for patients with TNBC who acquire RT-R and may improve their prognosis.

**Abstract:**

ESM-1, overexpressed in several cancer types, is a potential cancer diagnostic and prognostic indicator. In our previous study, we determined that RT-R-TNBC cells were more aggressive than TNBC cells, and this difference was associated with ESM-1 overexpression. However, the mechanism explaining upregulated ESM-1 expression in RT-R-TNBC cells compared to TNBC cells was unclear. Therefore, we aimed to identify the mechanism by which ESM-1 is overexpressed in RT-R-MDA-MB-231 cells. RT-R-MDA-MB-231 cells were treated with various ESM-1 transcription factor inhibitors, and only the FoxO1 inhibitor downregulated ESM-1 expression. FoxO1 nuclear localization was modulated by JNK and p38 MAPKs, which were differentially regulated by PKC, PDK1 and PAK1. PAK1 profoundly modulated JNK and p38 MAPKs, whereas PKC and PDK1 affected only p38 MAPK. P2Y_2_R activated by ATP, which is highly released from RT-R-BC cells, was involved in PAK1 activation, subsequent JNK and p38 MAPK activation, FoxO1 induction, and ESM-1 expression in RT-R-MDA-MB-231 cells. These findings suggest for the first time that ESM-1 was overexpressed in RT-R-MDA-MB-231 cells and regulated through the P2Y_2_R-PAK1-FoxO1 signaling pathway.

## 1. Introduction

A specific subtype of breast cancer known as triple-negative breast cancer (TNBC) has high mortality, numerous early relapses, and a dismal prognosis. TNBC is characterized as the absence of estrogen receptor (ER), progesterone receptor (PR), and human epidermal growth factor receptor-2 (HER-2) [1]. The main treatments for breast cancer are surgery, hormone therapy, chemotherapy, radiotherapy and targeted therapy, and one of these treatments or a combination is used depending on the method of diagnosis and stage of the cancer. Hormone therapy used to target ER and PR is suitable for ER- and PR-positive breast cancer patients [2]; however, patients with TNBC have fewer treatment options than patients with other types of breast cancer due to the absence of therapeutic targets such as ER, PER, and HER2 [3]. Therefore, TNBC patients are usually treated with chemotherapy (CT) and radiotherapy (RT) to shrink a large tumor before surgery or to destroy any remaining cancer cells after surgery. Unfortunately, TNBC cells are likely to acquire resistance to RT and, therefore, TNBC patients experience a higher rate of local relapse after RT than patients with ER-positive breast tumors. Preclinical evidence suggests that ER may be a modulator of radiosensitivity and that its absence may contribute to the RT resistance of TNBC cells [4,5]. Our previous study revealed that RT-resistant (RT-R)-TNBC cells exhibited highly aggressive features, including an increased proliferation rate, invasiveness, and metastatic capacity [6]. Once the disease recurs with the acquisition of RT resistance, it can easily spread to distant organs and contribute to cancer-related patient death. Thus, it is important to find an effective therapeutic target that is applicable to patients with RT-R-TNBC.

Endothelial cell specific molecule-1 (ESM-1), also known as endocan, is a 50-kDa endothelial cell-associated proteoglycan [7]. ESM-1 has been recently studied as a biomarker candidate for endothelial dysfunction and inflammation because increased ESM-1 levels in tissue or serum reflect endothelial activation and neoangiogenesis, which are marked pathophysiological changes involved in inflammation [8,9,10]. In addition, it has been reported that ESM-1 is expressed preferentially in the tumor endothelium [11] and is also overexpressed in several types of cancers, including gastric cancer, colorectal cancer, hepatocellular carcinoma, non-small lung cancer, ovarian cancer, and brain cancer [12,13,14,15,16,17]. Several lines of evidence suggest that ESM-1 may be a potential marker for cancer diagnosis and prognosis and a therapeutic target for cancer treatment; ESM-1 levels in the serum of patients with gastric cancer or colorectal cancer were found to be significantly higher than those in healthy volunteers, and the overall survival was reduced in gastric cancer or colorectal cancer patients with high levels of serum ESM-1 compared with those with low levels, indicating a correlation with poor prognosis [18,19]. Moreover, TNBC patients with high levels of ESM-1 expression in tumors showed notably worse recurrence-free survival than those with low ESM-1 expression [20]. Furthermore, in our previous studies, we established RT-R-breast cancer cells from TNBC and non-TNBC cells [6] and analyzed alterations in gene expression between MDA-MB-231 (TNBC) and RT-R-MDA-MB-231 (RT-R-TNBC) cells [21]. It is interesting that the ESM-1 was the most increased gene in RT-R-MDA-MB-231 cells compared to MDA-MB-231 cells. We previously discovered that ESM-1 played an important role in the tumorigenesis of breast cancer cells, especially RT-R-TNBC cells, through the regulation of several genes involved in tumor growth, progression, and metastasis in vitro and in vivo [21]. However, it is still unknown how ESM-1 is highly expressed in RT-R-TNBC cells. We therefore sought to reveal the mechanisms by which ESM-1 is overexpressed in RT-R-MDA-MB-231 cells compared to MDA-MB-231 cells in the present study and propose a new therapeutic strategy for the treatment of patients with RT-R-TNBC.

Recent reports have suggested that purine molecules, such as ATP and adenosine, are known to bind to various purinergic receptors that are expressed on the cell surface and act as signal transducers in diverse types of cells, including breast cancer cells [22]. In a previous study, we reported that, compared to MDA-MB-231 cells, RT-R-MDA-MB-231 cells secrete higher levels of ATP and adenosine [23], and ATP and adenosine promote tumor progression in TNBC cells and RT-R-TNBC cells through the activation of the P2Y_2_ receptor (P2Y_2_R) and A2A receptor (A2AR), respectively [23,24]. In addition, accumulating evidence supports the idea that the A2B receptor (A2BR), which is highly expressed in various tumors, plays an oncogenic role in many malignancies [25]. Therefore, we investigate whether the purinergic receptors that we have previously shown to increase the proliferation and metastasis of RT-R-TNBC cells may be associated with the upregulated expression of ESM-1 in RT-R-MDA-MB-231 cells.

## 2. Materials and Methods

### 2.1. Cell Culture and Establishment of RT-R-BC Cells

The human breast cancer cell line MDA-MB-231 was provided from Korea Cell Line Bank (Seoul, Korea). RT-R-MDA-MB-231 cells were established by repeatedly irradiating with fractionated X-rays (each 2 Gy, 25 fractions) until reaching a total of 50 Gy, as described previously [6]. These breast cancer cells were grown in RPMI-1640 medium (SH30027.01, Cytiva, Marlborough, MA, USA) supplemented with 10% fetal bovine serum (FBS; F0900, GenDEPOT, Katy, TX, USA) and 1% penicillin and streptomycin (SV30010, Cytiva) at 37 °C in humidified air containing 5% CO_2_. Cells were used only in the first 5 passages for experiments.

### 2.2. Reagents

Digoxin (4683), Bay11-7085 (1743), SR11302 (2476), AS1842856 (4265), AG490 (0414), SP600125 (1496), SB203580 (1202), GF109203X (0741) and OSU03012 (5682) were purchased from Tocris (Bristol, UK). PD98059 (ab146592) was obtained from Abcam (Cambridge, UK). FRAX1036 (SML2618), ATP (A2383) and apyrase (A6410) were purchased from Sigma-Aldrich (St. Louis, MO, USA).

### 2.3. Total RNA Extraction and Reverse Transcription-Polymerase Chain Reaction (RT–PCR)

Total RNA was extracted from breast cancer cells (1 × 10^6^ cells) using TRIzol reagent (15596018, Thermo Fisher Scientific, Waltham. MA, USA) according to the manufacturer’s protocol. RT–PCR was conducted using TOPscript One-step RT–PCR Drymix (RT421, Enzynomics, Daejeon, Korea) by the manufacturer’s instructions. Human ESM-1 and glyceraldehyde 3-phosphate dehydrogenase (GAPDH) primers were purchased from Bioneer (Deajeon, Korea), and the sequences of these primers were as follows: hESM-1 forward, 5′-GC CCT TCC TTG GTA GGT AGC-3′, and reverse, 5′-TG TTT CCT ATG CCC CAG AAC-3′, and hGAPDH forward, 5′- TCA ACA GCG ACA CCC ACT CC-3′, and reverse, 5′-TGA GGT CCA CCC TGT TG-3′. Amplification was conducted under the following conditions: 27 cycles of denaturation at 95 °C for 30 sec, annealing at 57.5 °C for 30 sec, and extension at 72 °C for 1 min.

### 2.4. Cell Viability Assay

RT-R-MDA-MB-231 (1 × 10^4^ cells) samples were seeded in 96-well plates, and then treated with AS1842856 in a dose-dependent manner (10, 50, 100 and 200 µM). After 24 h of treatment, 10 µL of D-Plus™ cell counting kit viability assay reagent (CCK-3000, Dongin Biotech, Seoul, Korea) was added to each well in the 96-well plates, and stained for 30 min in the dark at 37 °C. Then, the optical density of each well was measured at a wavelength of 450 nm with a microplate reader.

### 2.5. Protein Extraction from Whole-Cell Lysates or Nuclear/Cytosolic Fractions and Western Blot Analysis

Cells (5 × 10^6^ cells) were washed with ice-cold 1 × PBS and lysed with radioimmunoprecipitation assay (RIPA) buffer (0.1% sodium dodecyl sulfate (SDS) and 0.1% nonyl phenoxylpolyethoxylethanol-40 (NP-40) in 1 × PBS) containing 1% protease inhibitor cocktail for 1 h on ice. Then, the supernatant containing protein extract was obtained by centrifuging the suspension at 16,000× *g* for 15 min at 4 °C. The protein from the nuclear or cytosolic fraction was extracted with a nuclear/cytosol fractionation kit (K266, BioVision, Milpitas, CA, USA) according to the manufacturer’s instructions. Briefly, the cells were lysed with CEB-A mix containing DTT and protease inhibitors, and subsequently CEB-B was added before protein extraction. The suspension was centrifuged at 16,000× *g* for 5 min at 4 °C, and the supernatant was obtained as cytoplasmic extracts. The remaining pellet was resuspended in NEB mix containing DTT and protease inhibitors, and the supernatant (nuclear extract) was obtained by centrifuging the suspension at 16,000× *g* for 10 min at 4 °C. The protein obtained from the cells was applied to 8% SDS-polyacrylamide gel electrophoresis (PAGE) and then transferred onto polyvinylidene fluoride (PVDF) membranes. After blocking with 5% nonfat milk in TBS-T, the membranes were incubated with anti-ESM-1 (ab103590, Abcam), anti-Forkhead box O1 (FoxO1) (2880, Cell Signaling, Danvers, MA, USA), anti-phospho-c-Jun N-terminal kinase (JNK) (9251, Cell Signaling), anti-JNK (9252, Cell Signaling), anti-phospho-p38 (9211, Cell Signaling), anti-p38 (sc-535, Santa Cruz Biotechnology, Dallas, TX, USA), anti-phospho-protein kinase C (PKC) (9375, Cell Signaling), anti-PKC (sc-10800, Santa Cruz Biotechnology), anti-phospho-phosphoinositide-dependent kinase 1 (PDK1) (3061, Cell Signaling), anti-PDK1 (3062, Cell Signaling), anti-phospho-p21-activated kinase 1 (PAK1) (2601, Cell Signaling), anti-PAK1 (2602, Cell Signaling), anti-β-actin (MA5-15739, Thermo Fisher Scientific), and anti-proliferating cell nuclear antigen (PCNA) (sc-25280, Santa Cruz Biotechnology) antibodies. Bound antibodies were detected with horseradish peroxidase (HRP)-conjugated secondary antibodies and a Western ECL substrate (170-5061, Bio–Rad Laboratories Hercules, CA, USA).

### 2.6. Small Interfering RNA (siRNA) Transfection

Cells were transfected with 100 nM negative control siRNA or P2Y_2_R siRNA (Bioneer, Daejeon, Korea) using Lipofectamine 3000 (L300015, Thermo Fisher Scientific) in serum-free medium. After 6 h, the medium was replaced with fresh complete medium, and the cells were incubated for an additional 18 h at 37 °C. Then, the transfected cells were treated with indicated reagents.

### 2.7. Statistical Analysis

GraphPad Prism 7 was used to statistically analyze all the data. A one-way ANOVA and Tukey’s post hoc test were used to assess group differences. The results are presented as means ± standard deviation (SD).

## 3. Results

### 3.1. The FoxO1 Transcription Factor Is Involved in ESM-1 Overexpression in RT-R-MDA-MB-231 Cells

In our previous studies, we analyzed the genes between MDA-MB-231 cells and RT-R-MDA-MB-231, and found that ESM-1 was most highly upregulated gene in RT-R-MDA-MB-231 cells. Additionally, RT-R-MDA-MB-231 cells displayed higher expression levels of hypoxia-inducible factor (HIF-1α), activated nuclear factor-κB (NF-κB), and signal transducer and activator of transcription-3 (STAT-3) compared to MDA-MB-231 cells [6,21]. In addition, it has been reported that the transcription factors activator protein-1 (AP-1) and FoxO1 regulate the expression of ESM-1 as transcription factors [26,27,28,29,30,31]. Therefore, we sought to determine which transcription factors are involved in ESM-1 overexpression in RT-R-MDA-MB-231 cells. First, we confirmed that, compared to MDA-MB-231 cells, RT-R-MDA-MB-231 cells expressed higher levels of ESM-1 mRNA. Then, the RT-R-MDA-MB-231 cells were treated with various specific inhibitors against candidate transcription factors for 8 h, and we found that ESM-1 mRNA levels increased in the RT-R-MDA-MB-231 cells and were significantly downregulated only by treatment with AS1842856 (100 nM), a FoxO1 inhibitor (Figure 1A). Treatment with AS1842856 at 100 nM decreased both ESM-1 mRNA and protein levels in RT-R-MDA-MB-231 cells in a time-dependent manner and showed a maximum effect at 8 h for the mRNA level and at 8~16 h for the protein level (Figure 1B,C and Appendix A). In addition, ESM-1 mRNA and protein levels that were induced in RT-R-MDA-MB-231 cells at 8 h and 16 h, respectively, were decreased by AS1842856 (10, 50, 100 and 200 nM) in a dose-dependent manner (Figure 1D,E and Appendix A) without notable toxicity at these concentrations (Figure 1F). These results suggest that FoxO1 is a transcription factor that regulates ESM-1 overexpression in RT-R-TNBC cells.

### 3.2. Nuclear FoxO1 Levels Are Affected by JNK and P38 MAPK, but Not ERK, in RT-R-MDA-MB-231 Cells

Next, we investigated the upstream regulators that affect nuclear FoxO1 levels and thus regulate ESM-1 expression. Mitogen-activated protein kinases (MAPKs) are reported to be involved in FoxO activity and nuclear localization [32,33,34,35]. Therefore, we investigated the effect of JNK, P38, and extracellular signal-regulated kinase (ERK) MAPKs on nuclear FoxO1 levels in RT-R-MDA-MB-231 cells. When RT-R-MDA-MB-231 cells were treated with specific inhibitors against JNK, P38 and ERK MAPK activity, the inhibition of JNK and p38 activation led to a marked decrease in the FoxO1 level in the nucleus but an increase in the cytosol (Figure 2A,B and Appendix A). Interestingly, the inhibition of ERK activation did not lead to significant alterations in FoxO1 levels in either the cytosol or nucleus (Figure 2C and Appendix A). Similarly, ESM-1 mRNA and protein expression levels were notably downregulated in response to JNK and p38, but not the ERK inhibitor (Figure 2D,E and Appendix A). These results suggest that JNK and p38 MAPK are upstream regulators that affect nuclear FoxO1 levels to regulate ESM-1 expression in RT-R-TNBC cells.

### 3.3. PKC, PDK1 and PAK1 Differentially Regulate JNK and P38 MAPK-FoxO1-ESM-1 Cascades as Upstream Signaling Molecules in RT-R-MDA-MB-231 Cells

PKC, PDK1 and PAK1 are involved in the regulation of MAPK activity [36,37,38]. Therefore, we investigated whether PKC, PDK1 and PAK1 are upstream signaling molecules that regulate MAPK-FoxO1-ESM-1 cascades in RT-R-MDA-MB-231 cells. RT-R-MDA-MB-231 cells showed significantly increased JNK and p38 MAPK activity levels compared to those in the MDA-MB-231 cells (Figure 3A and Appendix A). The PKC inhibitor GF109203X (100 nM) and the PDK1 inhibitor OSU03012 (1 μM) effectively decreased p38 activation but not JNK activation (Figure 3B,C and Appendix A). Importantly, the inhibition of PAK1 using FRAX1036 (5 μM) dramatically inhibited both p38 and JNK activation (Figure 3D and Appendix A). Then, we wondered whether PKC, PDK1 and PAK1 affect the activation of each other. Figure 4A shows that levels of the active forms of PKC, PDK1 and PAK1 were increased in RT-R-MDA-MB-231 cells compared to MDA-MB-231 cells (Figure 4A and Appendix A). The inhibition of PKC did not modulate PDK1 or PAK1 activation (Figure 4B and Appendix A), and the PDK1 and PAK1 inhibitors did not affect PKC activation (Figure 4C,D and Appendix A). However, the PDK1 and PAK1 inhibitors affected PAK1 and PDK1 activation, respectively (Figure 4C,D and Appendix A).

Thereafter, we confirmed that PKC, PDK1 and PAK1 affected nuclear FoxO1 levels and ESM-1 expression in RT-R-TNBC cells as upstream signaling molecules. As expected, specific inhibitors against PKC, PDK1 and PAK1 markedly decreased FoxO1 levels in the nucleus of the RT-R-MDA-MB-231 cells (Figure 5A–C and Appendix A). Moreover, the mRNA and protein expression levels of ESM-1 were significantly reduced by inhibiting PKC, PDK1 and PAK1 in RT-R-MDA-MB-231 cells (Figure 5D,E and Appendix A). Taken together, these results suggest that PKC, PDK1 and PAK1 differentially regulated JNK and P38 MAPKs; PKC independently regulated p38 MAPK, whereas PDK1 and PAK1 modulated each other’s activity upstream of JNK and/or p38 MAPKs in the RT-R-MDA-MB-231 cells. In particular, PAK1 may play a critical role in regulating ESM-1 expression via the JNK and P38 MAPK-FoxO1 cascades in RT-R-TNBC cells.

### 3.4. P2Y_2_ Purinergic Receptor Activated by ATP Is Involved in ESM-1 Expression in RT-R-MDA-MB-231 Cells

Overexpressed ESM-1 is associated with the enhancement of tumorigenesis of RT-R-TNBC cells [21]. Moreover, RT-R-MDA-MB-231 cells release more ATP and adenosine extracellularly than MDA-MB-231 cells [23], and ATP and adenosine promote the tumor progression of RT-R-TNBC cells through the activation of P2Y_2_R and adenosine receptors (A2AR, A2BR, etc., respectively) [23,24,25]. Therefore, we further investigated the role of purinergic receptors on ESM-1 expression in RT-R-TNBC cells. Knocking down P2Y_2_R expression significantly reduced ESM-1 mRNA and protein levels in RT-R-TNBC cells (Figure 6A and Appendix A); however, A2AR and A2BR siRNA did not affect ESM-1 expression levels (Appendix A). In addition, P2Y_2_R activation by ATP treatment notably increased ESM-1 mRNA and protein levels, and this increase was abolished by knocking down P2Y_2_R expression (Figure 6B,C and Appendix A). Moreover, apyrase, a highly active ATP-diphosphohydrolase, slightly decreased the control level of ESM-1 mRNA and protein in RT-R-MDA-MB-231 cells (Figure 6B,C and Appendix A). Thus, these results suggest that P2Y_2_R activation by ATP plays an important role in the regulation of ESM-1 expression.

### 3.5. ATP-Activated P2Y_2_R Regulates ESM-1 Expression via the Activation of PAK1-JNK/p38 MAPK-FoxO1 Signaling Cascades in RT-R-MDA-MD-231 Cells

Lastly, we confirmed that ATP-activated P2Y_2_R regulates the signaling cascades associated with ESM-1 expression in RT-R-MDA-MB-231 cells. The knockdown of P2Y_2_R downregulated the increased level of FoxO1 in the nucleus of RT-R-MDA-MB-231 cells, instead increasing the cytosolic FoxO1 level (Figure 7A and Appendix A). Extracellular ATP treatment induced a nuclear FoxO1 level increase, which was inhibited by knocking down P2Y_2_R expression (Figure 7B and Appendix A). Removal of cellular ATP by apyrase decreased FoxO1 level in the nucleus of RT-R-MDA-MB-231 cells (Figure 7B and Appendix A). Furthermore, P2Y_2_R activation by ATP induced significant increases in JNK and p38 MAPK activation, as well as in PKC and PAK1 activation, and these increases were attenuated by knocking down P2Y_2_R expression in RT-R-MDA-MB-231 cells (Figure 7C,D and Appendix A). PDK1 activity was not regulated by ATP or mediated in a P2Y_2_R-dependent manner (Figure 7D and Appendix A).

## 4. Discussion

TNBC is a special subtype of breast cancer, accounting for 10–20% of breast cancer cases, and among the subtypes of breast cancer, TNBC has the worst prognosis, with no targeted therapy available. TNBC patients do not benefit from hormonal therapy due to the lack of ER, PR, and HER2 expression. Targeted therapy changes the way cells function and helps to suppress cancer growth and spreading. However, depending on the stage and type of breast cancer, not all breast cancer patients can be treated with targeted therapy [3]. Therefore, CT and RT are often applied to TNBC patients. However, CT and RT not only kill cancer cells but also influence normal and healthy cells, such as immune cells, breast skin cells, and normal cells surrounding cancer cells. Furthermore, when TNBC with naturally high malignancy acquires resistance against RT, it becomes more difficult to treat and to remove cancer cells than other breast cancers. Therefore, in our previous study, we established RT-R-TNBC cells and found that RT-R-TNBC cells exhibited more aggressive properties and promoted tumor progression more than TNBC cells [6,24]. Moreover, we revealed that ESM-1 showed the most increased expression level in RT-R-MDA-MB-231 cells compared to MDA-MDA-231 cells and suggested that ESM-1 overexpression in RT-R-MDA-MB-231 cells is associated with the higher aggressiveness of RT-R-TNBC cells compared to TNBC cells [21]. However, the mechanism by which ESM-1 expression is upregulated in RT-R-MDA-MB-231 cells has not yet been elucidated.

According to the reports of studies performed to date, the promoter region of ESM-1 has AP-1, HIF-1α and NF-κB binding sites, suggesting that AP-1, HIF-1α and NF-κB may be important regulators of ESM-1 expression [26,30]. In addition, ESM-1 expression has been reported to be enhanced by HIF-1α in response to hypoxia in human colorectal cancer [27] and to be mediated by NF-κB in IL-1β-induced inflammatory conditions in human chondrocytes [28]. Furthermore, another study reported that ESM-1 expression was regulated via the JAK/STAT3 pathway in EGFR-activated non-small-cell lung cancer cells [29] and induced by FoxO1 nuclear localization in cultured endothelial cells under hypoxic conditions [31]. Importantly, our previous studies showed that the level of HIF-1α and the activities of NF-κB and STAT-3 were induced and associated with tumor development in RT-R-MDA-MB-231 cells compared to MDA-MB-231 cells [6,21]. Hence, we first examined whether AP-1, HIF-1α, NF-κB and STAT-3 are involved in the expression of ESM-1 in RT-R-TNBC cells. However, when we measured the mRNA levels of ESM-1 in RT-R-MDA-MB-231 cells treated with specific inhibitors against the aforementioned transcription factors, ESM-1 mRNA levels were found to be overexpressed in RT-R-MDA-MB-231 cells compared to the levels in MDA-MB-231 cells, which were not affected by the inhibition of the four candidate transcription factors). Interestingly, only the specific inhibitor of FoxO1, which is a candidate ESM-1 transcription factor, decreased the ESM-1 mRNA levels that had been upregulated in RT-R-MDA-MB-231 cells.

Forkhead box O proteins (FoxOs) constitute a family of transcription factors that are characterized by a forkhead box (Fox) or a conserved winged-helix DNA-binding motif, and directly bind to various target sequences [39,40]. FoxOs activate or inhibit downstream target genes transcriptionally, thereby regulating cell proliferation, apoptosis, metabolism, differentiation, autophagy, inflammation, and stress resistance [41]. FoxOs have been generally considered tumor suppressors due to their inhibitory effects on cancer cell growth, survival and metastasis [42]. In contrast, FoxOs can support tumor progression by participating in metastasis formation, maintaining cellular redox homeostasis, inducing drug resistance, and mediating growth factor signaling feedback [43]. The FoxOs expressed in mammals are FoxO1, FoxO3, FoxO4, and FoxO6, and notably, FoxO1 and FoxO3 are expressed in most tissues [44]. Among FoxO members, FoxO1 plays important roles in cell dissemination and anticancer drug resistance in breast cancer due to its direct regulation of the transcription of the metastatic factors MMP-1 and the drug efflux pump multidrug resistance 1 (MDR1) [45,46]. Furthermore, as a transcription factor, FoxO1 has been reported to regulate ESM-1 expression [31]. However, the role played by FoxO1 in RT-R-TNBC cells has not yet been determined; in particular, the involvement of FoxO1 with ESM-1, which is highly increased in RT-R-MDA-MB-231 cells compared to MDA-MB-231 cells, was determined in this study. We found that FoxO1 plays an important role as a transcription factor regulating ESM-1 expression, and this effect is closely related to RT-R-TNBC cell progression. In addition, our results showed that in addition to nuclear FoxO1 level, the cytosolic FoxO1 level was increased in RT-R-MDA-MB-231 cells compared to these levels in MDA-MB-231 cells (Figure 1). From these results, we suggest that nuclear FoxO1 is an important transcription factor that induces ESM-1 expression; however, the increased cytosolic FoxO1 level in tumor cells may be correlated with tumor progression. The differential roles played by FoxO1 in the nucleus and cytosol in TNBC need to be studied further.

Next, we identified the signaling molecules that regulate FoxO1 and increase the expression of ESM-1 through upstream signaling in RT-R-MDA-MB-231 cells. FoxOs are integration points for signals emitted by various sources, and activated kinases in different pathways can either activate or inhibit FoxO transcriptional activity [47]. Among the various signaling molecules, MAPKs, including JNK, ERK and p38, have been reported to be related to FoxO activity and nuclear localization; for example, it was found that JNK, which is activated by oxidative stress, phosphorylated the 14-3-3 protein or antagonized AKT to enhance FoxO activity and promote FoxO protein nuclear localization [32,33]. In addition, JNK phosphorylated FoxO4 directly to induce FoxO4 nuclear localization under oxidative stress [48]. Moreover, ERK2 regulated the epithelial-to-mesenchymal transition (EMT) plasticity in breast epithelial cells through FoxO1 activation [34], and p38 activated by anticancer drugs affected colorectal cancer growth via the regulation of FoxO3 nuclear localization [35]. These findings support the hypothesis that MAPKs are associated with FoxO activity and nuclear localization; however, whether MAPKs can regulate FoxO1 activity or nuclear localization in RT-R-TNBC to increase ESM-1 expression has not been determined. Therefore, in this study, we investigated the effect of MAPKs on the nuclear localization of FoxO1, as well as on ESM-1 expression in RT-R-MDA-MB-231 cells. Interestingly, JNK and p38 MAPKs, but not ERK, were involved in FoxO1 nuclear localization and the resulting increase in ESM-1 expression in RT-R-TNBC cells. In addition, compared to MDA-MB-231 cells, RT-R-MDA-MB-231 cells showed higher levels of JNK and p38 MAPK activity, supporting the role played by JNK and p38 activation to increase ESM-1 expression in RT-R-TNBC cells.

Furthermore, JNK and p38 MAPKs, which are involved in the regulation of ESM-1 expression, are modulated by PKC and PDK1. According to several reports, PKC activator phorbol myristate acetate induced the phosphorylation of JNK and ERK MAPKs in a PKC-dependent mechanism in leukemia cells [49]. In addition, PDK1 regulated chondrocyte apoptosis via the p38 MAPK pathway [50] and ovarian cancer cell metastasis through the modulation of JNK signaling [51]. Similar to PKC and PDK1, PAK1 stimulated the MAPK pathway to mediate related gene expression [52]. PAK1 is the best-characterized member of the PAK family, and is involved in various signaling pathways related to innate response, barrier function, epithelial cell migration, and cell survival [53,54], and PAK1 has been found to be upregulated and activated in several types of human cancers including breast cancer, colon cancer, and ovarian cancer [55,56,57]. Increased PAK1 expression and activity in breast cancer is correlated with higher tumor grade and higher invasiveness [55]. Considering these reports, we examined whether PKC, PDK1 and PAK1 modulate the activities of JNK and p38 MAPKs to regulate FoxO1 nuclear localization and subsequent ESM-1 expression, and our results showed that PKC and PDK1 regulated p38 activation but not JNK activation in RT-R-TNBC cells; however, PAKs activated both the p38 and JNK pathways. Interestingly, we found that PKC independently regulated p38 MAPK activity, whereas PDK1 and PAK1 affected each other’s activity upstream of JNK and/or p38 MAPKs in the RT-R-MDA-MB-231 cells, suggesting that PAK1 plays a more important role in the upregulation of ESM-1 expression in RT-R-TNBC cells through the regulation of both JNK and p38 MAPKs, which are upstream regulators, and PDK1 activity.

ATP exists abundantly in the tumor microenvironment because it is released from not only cancer cells but also from surrounding cells. Extracellular ATP binds to the purinergic P2Y receptor on the cell surface and activates the intracellular signaling pathway related to tumor growth and progression [58]. Our previous study reported that RT-R-BC cells released higher levels of ATP than breast cancer cells in response to the same stimulant, and the released ATP promoted breast tumor growth and metastasis through the activation of P2Y_2_R in both MDA-MB-231 and RT-R-MDA-MB-231 cells [39,59]. In addition, abundant ATP in the tumor microenvironment is easily converted to adenosine by the ectonucleotidases CD39 and CD73, which mediate intracellular responses by binding and activating the purinergic P1 receptors [58]. Among the four subtypes of P1 receptors (A1R, A2AR, A2BR and A3R), A2AR and A2BR have been previously determined to play roles in tumor progression: A2BR, which is expressed on both immune and nonimmune cells, is related to adenosine-induced tumor cell migration, invasion and metastasis [60,61]. In contrast, A2AR, which is highly expressed on immune cells, exhibits adenosine-mediated anti-inflammatory and immune-suppressing capacity [62]. Previously, we determined that adenosine-activated A2AR promoted breast tumor progression and metastasis by inducing AKT-β-catenin pathway activation [23]. Therefore, in this study, we investigated whether purinergic receptors are associated with upregulated ESM-1 expression, and if associated, we wanted to identify the receptor involved in ESM-1 overexpression in RT-R-TNBC cells. We found that the P1 adenosine receptors A2AR and A2BR did not affect ESM-1 expression, but P2Y_2_R activation by ATP induced PKC and PAK1 activation, JNK and p38 MAPK activation, and ultimately FoxO1 nuclear accumulation and ESM-1 expression in a P2Y_2_R-dependent manner in RT-R-MDA-MB-231 cells, suggesting an association between P2Y_2_R activation and ESM-1 expression in RT-R-TNBC cells.

## 5. Conclusions

Taken together, our results demonstrate, for the first time, the expression mechanism of ESM-1, which is upregulated and involved in the tumor progression of RT-R-TNBC cells. Particularly, the relationship between ESM-1 and purinergic receptors, especially P2Y_2_R, was identified in this study. Our results suggest that high levels of ATP released by RT-R-MDA-MB-231 cells may contribute to tumor progression by inducing P2Y_2_R activation and stimulating the P2Y_2_R-PAK1-FoxO1 signaling cascade to promote ESM-1 expression (Figure 8). Our findings suggest that P2Y_2_R-ESM-1 may be a new therapeutic target for patients with TNBC who acquire RT-R, and may improve their prognosis.

## Figures and Tables

**Figure 1 cancers-14-04124-f001:**
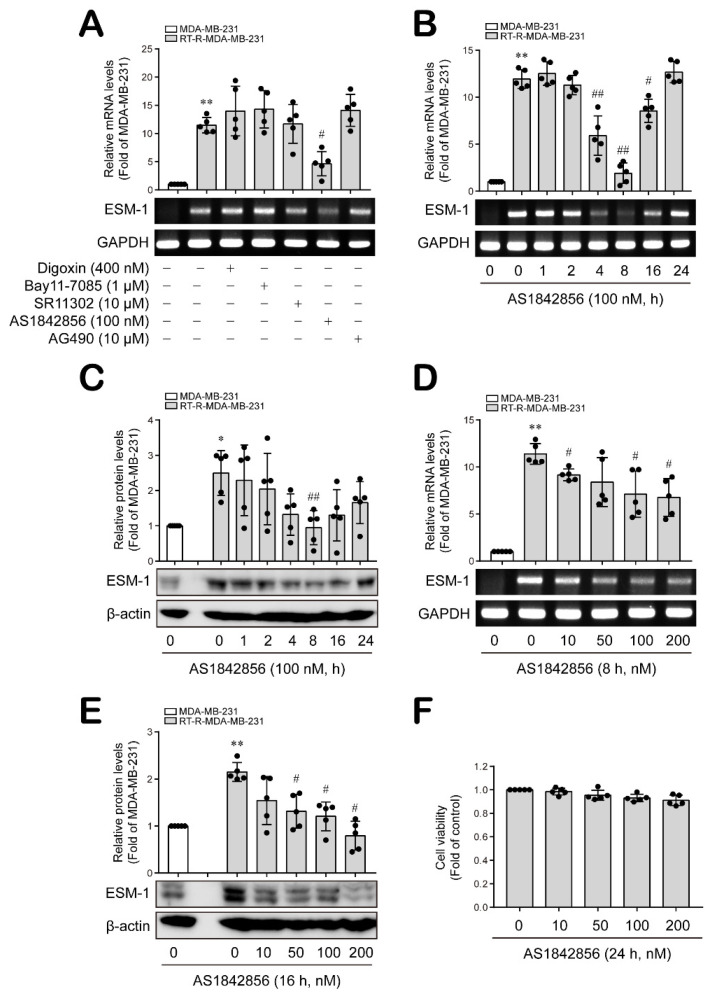
ESM-1 overexpression in RT-R-MDA-MB-231 cells compared to MDA-MB-231 cells was downregulated by an inhibitor of FoxO1. (**A**) MDA-MB-231 and RT-R-MDA-MB-231 cells were treated with the indicated inhibitors (digoxin, a HIF-1α inhibitor; Bay11-7085, an NF-κB inhibitor; SR11302, an AP-1 inhibitor; AS1842856, a FoxO1 inhibitor; and AG490, a STAT-3 inhibitor) for 8 h, and then ESM-1 and GAPDH mRNA expression levels were analyzed with total RNA extracted from the cells by RT–PCR. The results are presented as the mean ± SD of five independent experiments. ** *p* < 0.01 compared to the MDA-MB-231 cells; # *p* < 0.05 compared to the untreated RT-R-MDA-MB-231 cells. (**B**–**E**) MDA-MB-231 and RT-R-MDA-MB-231 cells were treated with AS1842856, and ESM-1 mRNA expression levels (**B**,**D**) and protein expression levels (**C**,**E**) were analyzed by RT-PCR and Western blotting, respectively. GAPDH and β-actin were used as loading controls. The results are presented as the mean ± SD of five independent experiments. * *p* < 0.05, ** *p* < 0.01 compared to the MDA-MB-231 cells; # *p* < 0.05, ## *p* < 0.01 compared to the untreated RT-R-MDA-MB-231 cells. (**F**) RT-R-MDA-MB-231 cells were treated with the indicated doses of AS1842856 for 24 h, and cell viability was measured by cell counting kit (CCK) viability assay. The results are presented as the mean ± SD of five independent experiments.

**Figure 2 cancers-14-04124-f002:**
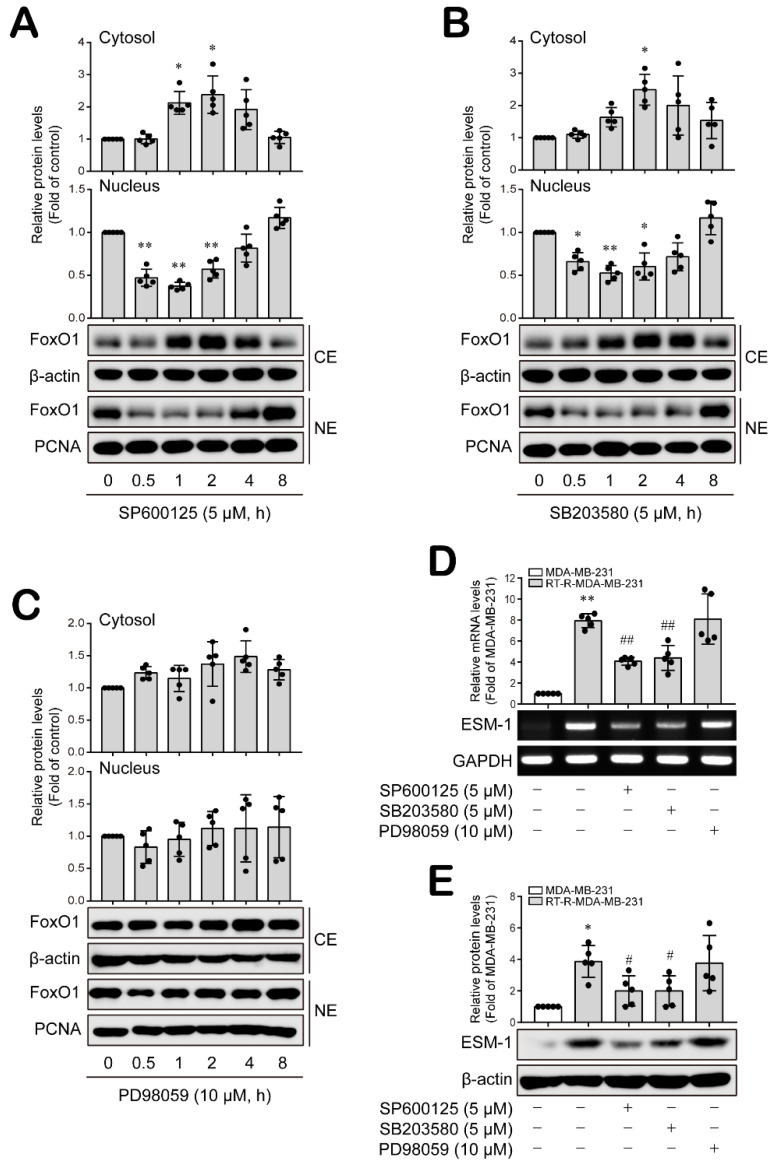
Nuclear FoxO1 levels and ESM-1 induction were mediated by JNK and P38 MAPK, but not ERK, in RT-R-TNBC cells. (A–C) RT-R-MDA-MB-231 cells were treated with SP600125 (5 μM; a JNK inhibitor) (**A**), SB203580 (5 μM; a p38 inhibitor) (**B**) and PD98059 (10 μM; an ERK inhibitor) (**C**) for the indicated times, and nuclear/cytosolic protein was extracted from the cells. FoxO1 protein levels in the cytosol or nucleus were analyzed by Western blotting. β-Actin and PCNA were used as loading controls. The results are presented as the mean ± SD of five independent experiments. * *p* < 0.05, ** *p* < 0.01 compared to the control. CE, cytosolic extract; NE, nuclear extract. (**D**,**E**) MDA-MB-231 and RT-R-MDA-MB-231 cells were treated with the indicated inhibitors for 8 h or 16 h, and then total RNA or protein was collected from the cells. ESM-1 and GAPDH mRNA levels were analyzed by RT–PCR (**D**). ESM-1 and β-actin protein levels were analyzed by Western blotting (**E**). The values are presented as the mean ± SD of five independent experiments. * *p* < 0.05, ** *p* < 0.01 compared to the MDA-MB-231 cells; # *p* < 0.05, ## *p* < 0.01 compared to the untreated RT-R-MDA-MB-231 cells.

**Figure 3 cancers-14-04124-f003:**
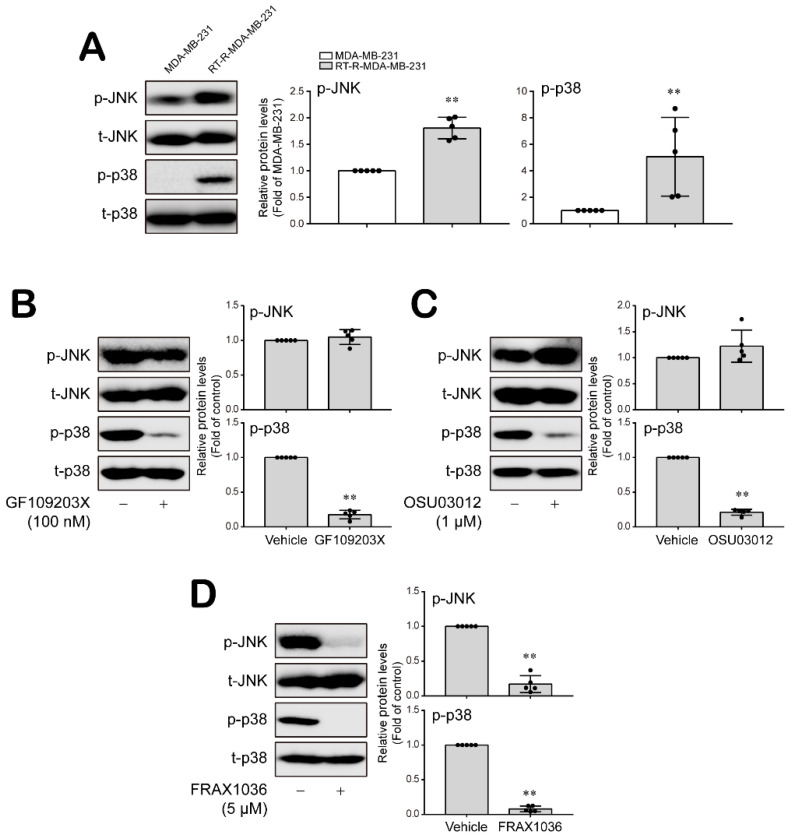
JNK and p38 MAPKs were differentially regulated by PKC, PDK1 and PAK1 in RT-R-MDA-MB-231 cells. (**A**) Phospho- and total JNK or p38 protein expression levels were analyzed with extracted proteins from MDA-MD-231 and RT-R-MDA-MB-231 cells by Western blotting. The results are presented as the mean ± SD of five independent experiments. ** *p* < 0.01 compared to the MDA-MB-231 cells. (**B**–**D**) RT-R-MDA-MB-231 cells were treated with GF109230X (100 nM; a PKC inhibitor, for 15 min), OSU03012 (1 μM; a PDK1 inhibitor, for 30 min) and FRAX1036 (5 μM; a PAK1 inhibitor, for 10 min), and then protein was collected from the cells. Phospho- and total JNK or p38 protein levels were analyzed by Western blotting. The values are presented as the mean ± SD of five independent experiments. ** *p* < 0.01 compared to the control.

**Figure 4 cancers-14-04124-f004:**
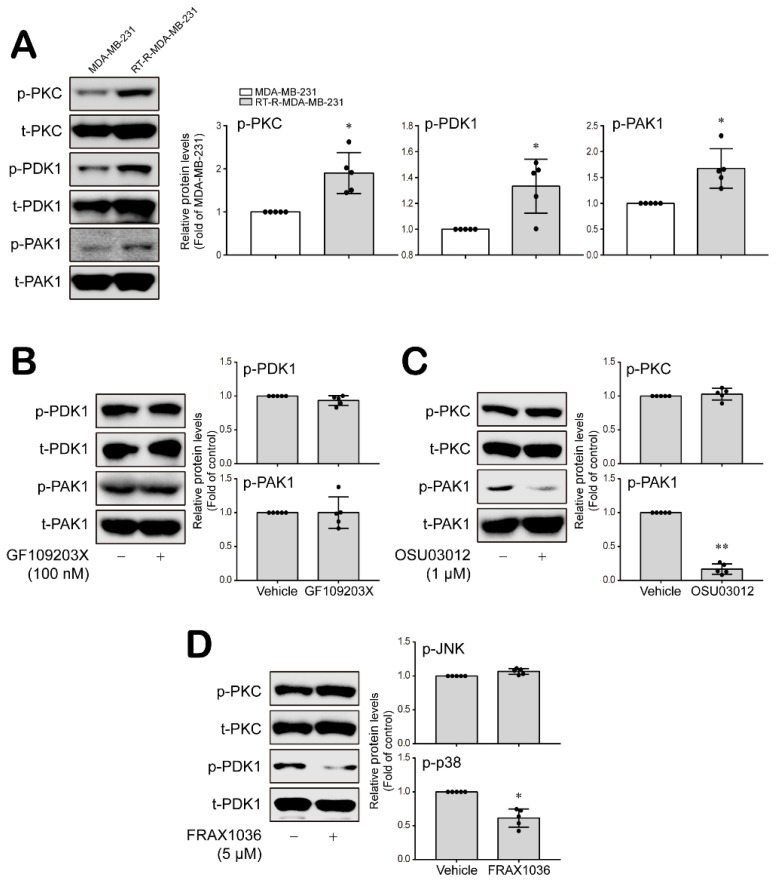
PDK1 and PAK1, which were activated in RT-R-MDA-MB-231 cells, interacted with each other but not with PKC. (**A**) Proteins were collected from MDA-MB-231 and RT-R-MDA-MB-231 cells, and phospho- and total-PKC, PDK1 and PAK1 protein levels were analyzed by Western blotting. The values are presented as the mean ± SD of five independent experiments. * *p* < 0.05 compared to the MDA-MB-231 cells. (**B**–**D**) RT-R-MDA-MB-231 cells were treated with GF109230X (100 nM) for 15 min, OSU03012 (1 μM) for 30 min, and FRAX1036 (5 μM) for 10 min. Total protein was extracted from the cells, and then phospho- and total-PDK1, PAK1 and PKC protein levels were analyzed by Western blotting. The values are presented as the mean ± SD of five independent experiments. * *p* < 0.05, ** *p* < 0.01 compared to the control.

**Figure 5 cancers-14-04124-f005:**
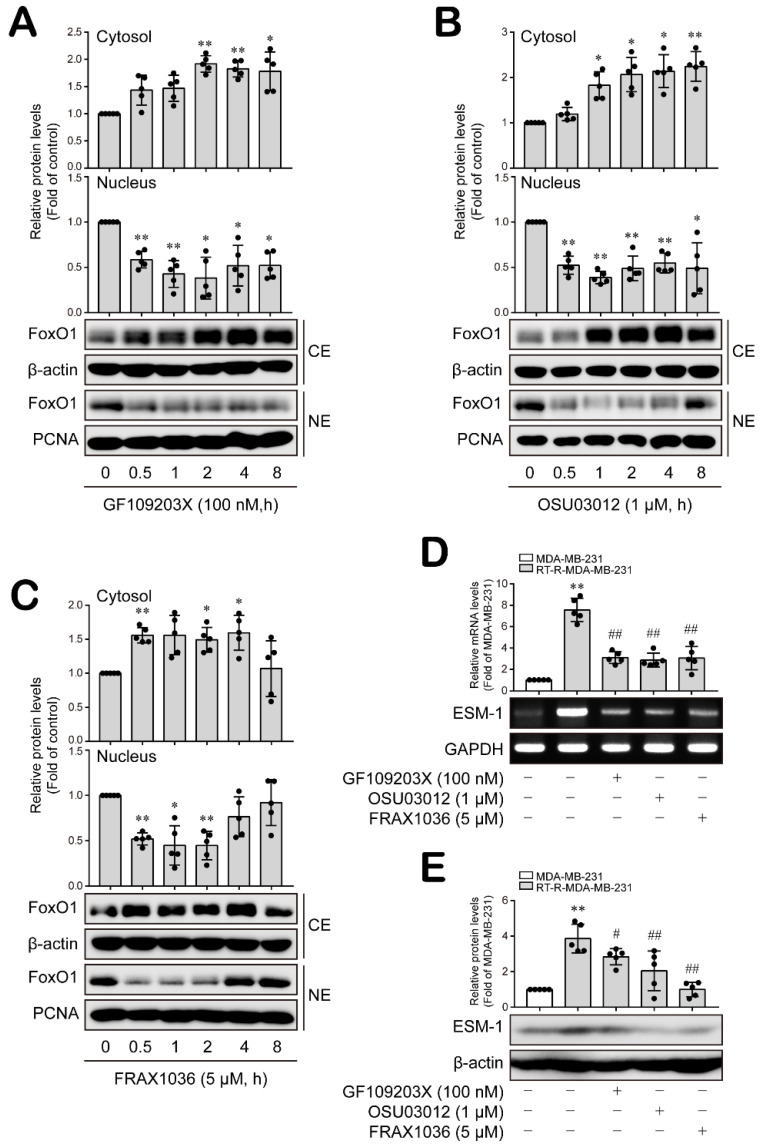
Nuclear FoxO1 levels and ESM-1 expression were regulated by PKC, PDK1 and PAK1 in RT-R-MDA-MB-231 cells. (**A**–**C**) RT-R-MDA-MB-231 cells were treated with GF109230X (100 nM), OSU03012 (1 μM) and FRAX1036 (5 μM) for the indicated times, and nuclear/cytosolic protein was extracted from the cells. FoxO1 protein expression levels in the cytosol or nucleus were analyzed by Western blotting. β-Actin and PCNA were used as loading controls. The values are presented as the mean ± SD of five independent experiments. * *p* < 0.05, ** *p* < 0.01 compared to the control. CE, cytosolic extract; NE, nuclear extract. (**D**,**E**) MDA-MB-231 and RT-R-MDA-MB-231 cells were treated with the indicated inhibitors for 8 h and 16 h before collection of total RNA and protein, respectively. ESM-1 and GAPDH mRNA levels were analyzed by RT–PCR. ESM-1 and β-actin protein levels were analyzed by Western blotting. The values are presented as the mean ± SD of five independent experiments. ** *p* < 0.01 compared to the MDA-MB-231 cells; # *p* < 0.05, ## *p* < 0.01 compared to the untreated RT-R-MDA-MB-231 cells.

**Figure 6 cancers-14-04124-f006:**
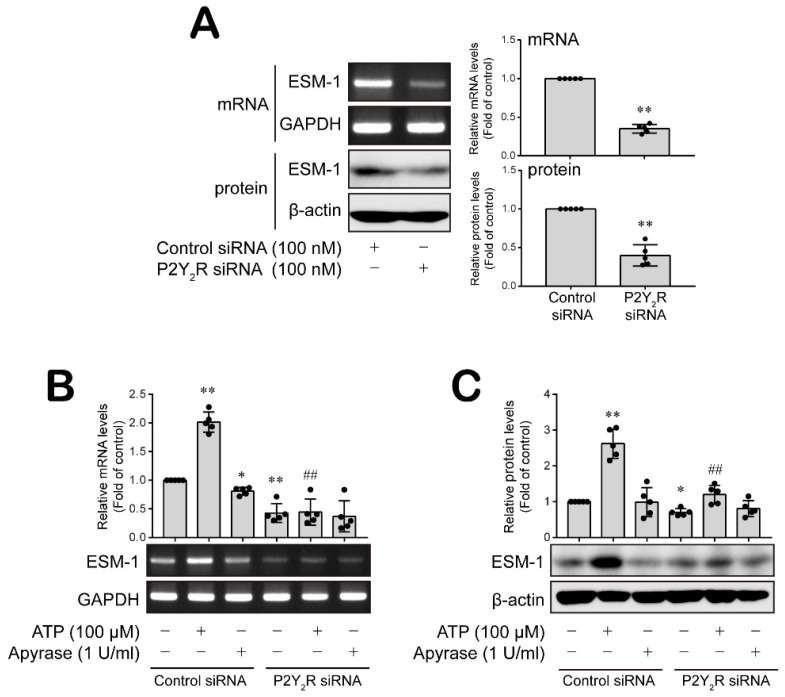
ATP-activated P2Y_2_R was involved in ESM-1 expression in RT-R-MDA-MB-231 cells. (**A**) After the efficiency of P2Y_2_R siRNA transfection was confirmed in RNA and protein level by RT-PCR and Western blotting, respectively (**A**), control or P2Y_2_R siRNA-transfected RT-R-MDA-MB-231 cells were treated with ATP (100 μM) or apyrase (1 U/mL; a highly active ATP-diphosphohydrolase) for 24 h (**B**,**C**). Total RNA and protein were collected from the cells, and ESM-1 mRNA (**B**) and protein (**C**) expression levels were analyzed by RT-PCR and Western blotting, respectively. GAPDH and β-actin were used as loading controls. The results are presented as the mean ± SD of five independent experiments. * *p* < 0.05, ** *p* < 0.01 compared to the control group in the control siRNA-transfected cells; ## *p* < 0.01 compared to the ATP-treated group in the control siRNA-transfected cells.

**Figure 7 cancers-14-04124-f007:**
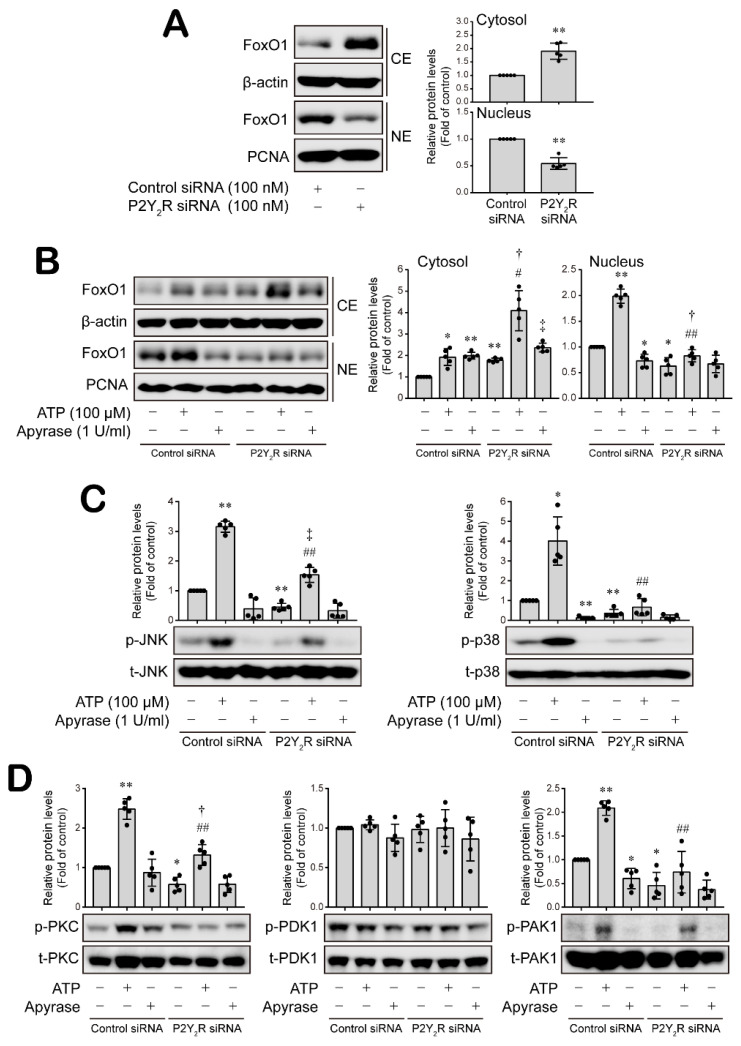
ATP-induced activation of P2Y_2_R regulated ESM-1 expression through the PAK1-JNK/p38 MAPK-FoxO1 signaling cascades in RT-R-MDA-MD-231 cells. (**A**) Nuclear/cytosolic protein was extracted from control or P2Y_2_R short interfering RNA (siRNA)-transfected RT-R-MDA-MB-231 cells, and FoxO1 protein levels in the cytosol and nucleus were analyzed by Western blotting. β-Actin and PCNA were used as loading controls. The values are presented as the mean ± SD of five independent experiments. ** *p* < 0.01 compared to the control siRNA-transfected cells. (**B**) After control or P2Y_2_R, siRNA-transfected RT-R-MDA-MB-231 cells were treated with ATP or apyrase for 8 h, and FoxO1 protein levels in the cytosol and nucleus were analyzed by Western blotting. (**C**) Control or P2Y_2_R siRNA-transfected RT-R-MDA-MB-231 cells were treated with ATP or apyrase for 5 min to detect JNK and for 60 min to detect p38, respectively. Phospho- and total JNK or p38 protein expression levels were analyzed from total protein fraction of cells by Western blotting. (**D**) Control or P2Y_2_R siRNA-transfected RT-R-MDA-MB-231 cells treated with ATP or apyrase were then treated for 5 min, and phospho- and total PKC, PDK1, or PAK1 protein levels were detected by Western blotting. The values are presented as the mean ± SD of five independent experiments. * *p* < 0.05, ** *p* < 0.01 compared to the control group in the control siRNA-transfected cells; # *p* < 0.05, ## *p* < 0.01 compared to the ATP-treated group in the control siRNA-transfected cells; † *p* < 0.05, ‡ *p* < 0.01 compared the control group in the P2Y_2_R siRNA-transfected cells. CE, cytosolic extract; NE, nuclear extract.

**Figure 8 cancers-14-04124-f008:**
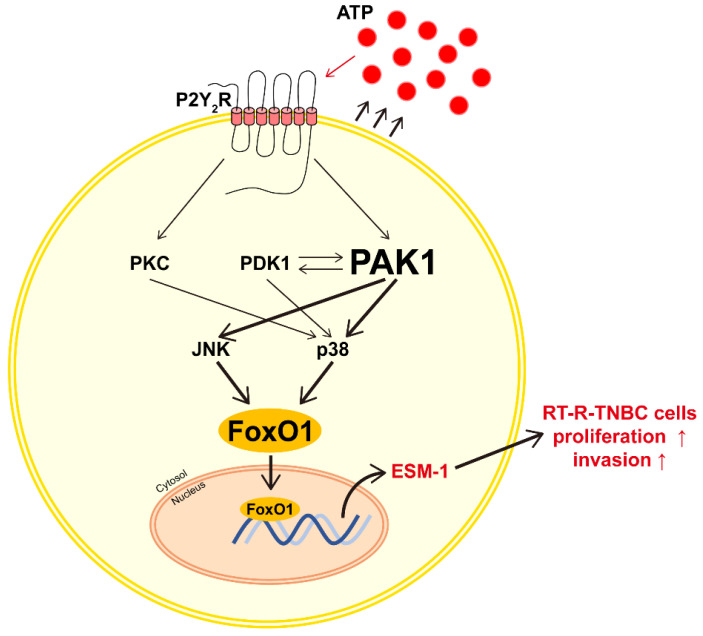
Schematic representation of the proposed mechanism of ESM-1 expression in RT-R-TNBC cells.

## Data Availability

The data presented in this study are available on request from the corresponding author.

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
