# Peer review of "P2Y2R-Mediated PAK1 Activation Is Involved in ESM-1 Overexpression in RT-R-MDA-MB-231 through FoxO1 Regulation"

_cancers, 2022, doi:10.3390/cancers14174124_

Round 1
Reviewer 1 Report
The manuscript entitled, “P2Y2R mediated PAK1 activation is involved in the ESM-1 overexpression in RT-R-MDA-MB-231 through FoxO1 regulation” is interesting. In this study, the authors have determined the mechanisms by which endothelial cell-specific molecule-1 (ESM-1) overexpression is regulated in radiotherapy-resistant (RT-R) triple-negative breast cancer (TNBC) cells. Overall, the studies have demonstrated that P2Y2R-PAK1-FoxO1 signaling regulates the overexpression of ESM-1 in RT-R-MDA-MB-231 cells, indicating the potential of this pathway as a new therapeutic target for the treatment of radiotherapy-resistant TNBC. The studies are nicely designed and executed. There are a few comments that need to be addressed.
Major comments:
1. In Figure 1, the authors have shown that the overexpression of ESM-1 mRNA levels was significantly attenuated by a FoxO1 inhibitor, AS1842586, but not by HIF-1a, NF-kB, STAT-3, and AP1 inhibitors in RT-R-MDA-MB-231 cells as compared to the parental cells. However, it is not clear why they used a 24 hours time point to determine the dose-dependent effect of AS1842586 when ESM-1 mRNA and protein levels were found to be significantly reduced at a 4-hour time point with much greater effect at 8 hours, and that at 24 hours’ time point, there is no effect of AS1842856 on ESM-1 levels?
Minor comments:
1. Please include the number of cells used for each assay (e.g., Cell viability assay, protein extraction from whole-cell, nuclear/cytosolic fractions, etc.) in the methods section.
2. The title of the result section 3.5 and Figure 7 legend is the same. Please consider revising one of them.
3. The authors mentioned “Not applicable” to the Data Availability Statement”. This should be revised.
Author Response
Major comments:
1. In Figure 1, the authors have shown that the overexpression of ESM-1 mRNA levels was significantly attenuated by a FoxO1 inhibitor, AS1842586, but not by HIF-1a, NF-kB, STAT-3, and AP1 inhibitors in RT-R-MDA-MB-231 cells as compared to the parental cells. However, it is not clear why they used a 24 hours time point to determine the dose-dependent effect of AS1842586 when ESM-1 mRNA and protein levels were found to be significantly reduced at a 4-hour time point with much greater effect at 8 hours, and that at 24 hours’ time point, there is no effect of AS1842856 on ESM-1 levels?
→ Answer: Thank you for your valuable comments. As you mentioned, the expression of ESM-1 was significantly reduced by AS1842586 treatment at 8 h in case of mRNA and at 8 ~ 16 h in case of protein in RT-R-MDA-MB-231 cells. There has been an error in Figure 1E. In Section 3.1, the valid treatment durations was demonstrated the same as above, and Figure 1 was corrected. Please check Figure 1.
Additionally, Figure 1F explains that the treatment of AS1842586 for 24 h did not exhibit any cytotoxicity in RT-R-MDA-MB-231 cells.
Minor comments:
1. Please include the number of cells used for each assay (e.g., Cell viability assay, protein extraction from whole-cell, nuclear/cytosolic fractions, etc.) in the methods section.
→ Answer: As you asked, we included the number of cells in some part of the Method section (2.3., 2.4., 2.5.). Please check the sections. Thank you.
2. The title of the result section 3.5 and Figure 7 legend is the same. Please consider revising one of them.
→ Answer: Thank you for your comment. We changed the title of Figure 7 legend, according to your suggestion.
3. The authors mentioned “Not applicable” to the Data Availability Statement”. This should be revised.
→ Answer: We have revised the Data Availability Statement as follows, “Data Availability Statement: The data presented in this study are available on request from the corresponding author.” (line 538).
Reviewer 2 Report
In this study, Jin and Kim investigated the mechanism of regulation for endothelial cell- specific molecule-1 (ESM-1) overexpression in a radiotherapy-resistant (RT-R) triple-negative breast cancer (TNBC) cell line, namely RT-R-MDA-MB-231. By analysis of mRNA and protein expression in response to numerous pharmalogical inhibitors, the authors delineated the P2Y2R-PAK1-JNK/p38MAPK-FoxO1 signaling pathway in regulating ESM-1 overexpression in RT-R-MDA-MB-231 cell cultures.
Overall, the manuscript is well written, and the results are interesting, well presented, and convincing.
Specific comments:
1. The key/legend for RT-MDA-MB-231 and RT-R-MDA-MB-231 samples on the graphs in several figures (Fig 1A-E, 2D-E, 3A, 4A, 5D-E) is too small to be legible, and thus the font size should be increased.
2. Figure callouts should not be used in the Discussion (except for when a new figure of a schematic/model is presented), so these should be removed.
Author Response
Specific comments:
1. The key/legend for RT-MDA-MB-231 and RT-R-MDA-MB-231 samples on the graphs in several figures (Fig 1A-E, 2D-E, 3A, 4A, 5D-E) is too small to be legible, and thus the font size should be increased.
→ Answer: Thank you for your valuable comments. As you mentioned, we increased the font size of the difficult-to-recognize key/legend for MDA-MB-231 and RT-R-MDA-MB-231, and revised the Figures. Please check Figure 1~5.
2. Figure callouts should not be used in the Discussion (except for when a new figure of a schematic/model is presented), so these should be removed.
→ Answer: As you pointed out, all figure callouts in the Discussion section have been removed. Thanks for the comment.